# IPO: An Improved Parrot Optimizer for Global Optimization and Multilayer Perceptron Classification Problems

**DOI:** 10.3390/biomimetics10060358

**Published:** 2025-06-02

**Authors:** Fang Li, Congteng Dai, Abdelazim G. Hussien, Rong Zheng

**Affiliations:** 1School of Humanities, Minnan Science and Technology College, Quanzhou 362332, China; 2College of Foreign Languages, Fujian Normal University, Fuzhou 350007, China; daicongteng@mku.edu.cn; 3Department of Computer and Information Science, Linköping University, 58183 Linköping, Sweden; aga08@fayoum.edu.eg; 4Faculty of Science, Fayoum University, Faiyum 63514, Egypt; 5Applied Science Research Center, Applied Science Private University, Amman 11831, Jordan; 6New Engineering Industry College, Putian University, Putian 351100, China; zro@ptu.edu.cn

**Keywords:** Parrot Optimizer, global optimization, multilayer perceptron, roulette fitness–distance balance, oral English teaching quality evaluation

## Abstract

The Parrot Optimizer (PO) is a new optimization algorithm based on the behaviors of trained Pyrrhura Molinae parrots. In this paper, an improved PO (IPO) is proposed for solving global optimization problems and training the multilayer perceptron. The basic PO is enhanced by using three improvements, which are aerial search strategy, modified staying behavior, and improved communicating behavior. The aerial search strategy is derived from Arctic Puffin Optimization and is employed to enhance the exploration ability of PO. The staying behavior and communicating behavior of PO are modified using random movement and roulette fitness–distance balance selection methods to achieve a better balance between exploration and exploitation. To evaluate the optimization performance of the proposed IPO, twelve CEC2022 test functions and five standard classification datasets are selected for the experimental tests. The results between IPO and the other six well-known optimization algorithms show that IPO has superior performance for solving complex global optimization problems. The results between IPO and the other six well-known optimization algorithms show that IPO has superior performance for solving complex global optimization problems. In addition, IPO has been applied to optimize a multilayer perceptron model for classifying the oral English teaching quality evaluation dataset. An MLP model with a 10-21-3 structure is constructed for the classification of evaluation outcomes. The results show that IPO-MLP outperforms other algorithms with the highest classification accuracy of 88.33%, which proves the effectiveness of the developed method.

## 1. Introduction

Metaheuristic algorithms are one type of stochastic searching method used to find an optimal solution within a given search space. In recent decades, metaheuristic algorithms have become very popular for solving different types of optimization problems due to their flexibility, no-derivation mechanism, and simplicity [1]. The exploration and exploitation phases are two important processes for the metaheuristic algorithms. The former is used to find the optimal solution in the global scope and avoid the local optimum, while the latter is used to improve the accuracy of the optimal solution. Up to now, researchers have proposed various metaheuristic algorithms according to the natural biological habits, physical and chemical laws, human behaviors and so on, such as Particle Swarm Optimization (PSO) [2], Grey Wolf Optimizer (GWO) [3], Snake Optimizer (SO) [4], Arithmetic Optimization Algorithm (AOA) [5], Reptile Search Algorithm (RSA) [6], Slime Mould Algorithm (SMA) [7], Remora Optimization Algorithm (ROA) [8], and recently proposed algorithms Pied Kingfisher Optimizer (PKO) [9], and Secretary Bird Optimization Algorithm (SBOA) [10].

One of the most important applications of metaheuristic algorithms is the training of artificial neural networks (ANNs). Among different types of artificial neural networks, the multilayer perceptron (MLP) has a simple structure and efficient performance and can be used to solve various classification problems in reality. The parameters, such as weights and biases of the MLP, can be optimized by the metaheuristic algorithms. Although there are many metaheuristic algorithms developed to train the MLP, according to the NFL theorem [11], new algorithms are always required when solving emerging complex optimization problems.

In the field of college oral English teaching, the goal is to improve students’ spoken language skills and develop their abilities to express their views clearly, accurately, and fluently, which are essential for their academic success, career development, and lifelong learning [12]. Strengthening supervision and feedback in the teaching process is a powerful means to ensure the teaching effect. Generally speaking, an integrated teaching quality evaluation system includes information from a supervisor, colleagues, students, and teachers. Therefore, the teaching quality evaluation model needs to consider a variety of factors, which can be a challenging nonlinear optimization problem. The traditional English teaching quality evaluation methods, such as the grey relational analysis method [13], the analytic hierarchy process [14], and fuzzy comprehensive evaluation method [15], are subjective and contain random defects, resulting in inaccurate evaluation results.

In the literature, there are some studies that have discussed teaching quality evaluations. For example, Zhang et al. [13] adopted the principal component analysis and support vector machine to improve the evaluation precision of English teaching quality. Lu et al. [16] constructed the English interpretation teaching quality evaluation model using the RBF neural network, which is optimized by a genetic algorithm. Wei et al. [17] investigated the evaluation performance of the college English teaching effect using an improved quantum particle swarm algorithm and a support vector machine. Zhang [18] also studied the problem of college English teaching effect evaluation, and the least squares support vector machine method was applied for the evaluation. Tan et al. [19] introduced the oral English teaching quality evaluation method based on a BP neural network, which is optimized by an improved crow search algorithm. Miao et al. [20] applied the decision tree algorithm to evaluate the teaching effect of oral English teaching with high accuracy and short time-consuming. Up to now, it still can be a challenge to develop an effective and reliable model for the accurate evaluation in the field of English teaching quality. The related works of English teaching quality evaluation are reported in Table 1.

Inspired by the four different behavioral characteristics of Pyrrhura Molinae parrots, Lian et al. proposed a new metaheuristic algorithm called Parrot Optimizer (PO) in 2024 [21]. The PO simulates parrots’ behaviors of foraging, staying, communicating, and fear of strangers, aiming to achieve a balance between exploration and exploitation. Although the PO algorithm produces satisfactory results for a variety of real-world engineering optimization problems, it does not perform well in solving high-dimensional optimization problems. In this paper, the PO algorithm is improved by using multiple improvement mechanisms and applied to solve the MLP classification problems.

The main contributions of this article are articulated as follows:An improved PO (IPO) is proposed in this paper, which adopts three improvements, namely the aerial search strategy, modified staying behavior, and communicating behavior;The proposed IPO is tested using twelve CEC2022 test functions;The numerical results, Wilcoxon signed-rank test, Friedman ranking test, convergence curves, and boxplots demonstrate the superiority of IPO compared to PO and the other five methods;The effectiveness of IPO-MLP is verified in training the multilayer perceptron for solving the classification problems, including five classification datasets and an oral English teaching quality evaluation problem.

The framework of the rest paper is outlined as follows: In Section 2, the methodology employed in this research is provided in detail. Section 3 gives the improvement methods of PO. The results of the proposed IPO on CEC2022, standard classification datasets, and the oral English teaching quality evaluation problem are presented in Section 4. Finally, Section 5 concludes this paper.

## 2. Preliminaries

### 2.1. Multilayer Perceptron

Multilayer perceptron (MLP) is one type of feedforward neural network (FNN), which has shown the powerful ability to solve nonlinear problems [22]. As shown in Figure 1, two MLP models are presented, which have structures of 3-4-3 and 3-4-4-3, respectively. The input information starts at the input layer and passes layer by layer until it reaches the output layer. In general, there is one input layer, one output layer, and one or more hidden layers in an MLP model, and each layer has multiple nodes. These nodes in the hidden or output layers are used to perform the neural network computation. Like other neural networks, the weights between nodes and the biases of nodes are the key parameters that need to be optimized.

The MLP calculation process can be represented by the following equations.

The first step is to calculate the weighted sum *H_j_* of *j*-th node by using Equation (1):(1)$$H_{j}=\sum_{i=1}^{m}(\omega_{ij}\times x_{i})-\theta_{j}$$
where *ω_ij_*, *x_i,_* and *θ_j_* are the weight, input, and bias of the *j*-th node, respectively.

Then the output of nodes is calculated by using Equation (2):(2)$$f(H_{j})=\frac{1}{1+\exp(-H_{j})}$$
where *f*(*H_j_*) denotes the active function calculation, and the applied function in this paper is the sigmoid function. In the next layer or output layer, the outputs are calculated in a manner similar to Equations (1) and (2). And finally, the output results are obtained.

For the training of MLP, the mean square error (MSE) is treated as the objective function, which is defined as follows:(3)$$\bar{MSE}=\sum_{n=1}^{S}\frac{\sum_{k=1}^{M}\left(e_{n}^{k}-o_{n}^{k}\right)}{S}$$
where *S* is the number of all samples. *M* is the number of nodes in the output layer. $e_{n}^{k}$ and $o_{n}^{k}$ indicate the output value and real value of the *k*-th output node for the *n*-th sample. According to Equation (3), the smaller the mean square error is, the closer the model output value is to the actual value. Therefore, the training process of the MLP model can be regarded as a minimum value problem, and the goal is to find the optimal weights and biases.

### 2.2. The Parrot Optimizer

The PO is a novel stochastic optimization algorithm inspired by the special behaviors of Pyrrhura Molinae parrots [21]. Four types of behavioral traits are modeled in PO: foraging, staying, communicating, and fear of strangers. The process of PO is described as follows.

#### 2.2.1. Population Initialization

In PO, the first step is to initialize the population. The position of the *i*-th agent is calculated as follows:(4)$$\vec{X_{i}^{0}}=rand\times \left(ub-lb\right)+lb,\;i=1,\;2,\;3,\;\ldots ,\;N$$
where *lb* and *ub* are the lower and upper boundaries, respectively; *rand* is a random value evenly distributed between 0 and 1. *N* is the size of the population. By using Equation (1), the initial positions of population individuals are randomly generated within the upper and lower boundary ranges.

#### 2.2.2. Foraging Behavior

In the foraging behavior, Pyrrhura Molinae parrots will consider the location of food or the owner and then fly to the estimated location. The mathematical model is described as follows:(5)$$X_{i}^{t+1}=\left(X_{i}^{t}-X_{best}\right)\times levy\left(D\right)+rand\times {\left(1-\frac{t}{T}\right)}^{\frac{2t}{T}}\times X_{mean}^{t}$$
where $X_{i}^{t+1}$ and $X_{i}^{t}$ indicate the positions of the *i*-th agent at the current and next iteration; *X_best_* denotes the location of food or the owner; *levy*(*D*) denotes the Levy distribution operator and *D* is the dimension of the objective function; $X_{mean}^{t}$ is the average location of the population.

#### 2.2.3. Staying Behavior

In the staying behavior, Pyrrhura Molinae parrots will fly to the owner and stay on the owner’s body for a while. This behavior is formulated as follows:(6)$$X_{i}^{t+1}=X_{i}^{t}+X_{best}\times levy\left(D\right)+rand\times ones\left(1,\;D\right)$$
where *ones*(1, *D*) represents an all-1 vector of *D* dimension. *X_best_* × *levy*(*D*) represents the behavior of flying to the owner, and *rand* × *ones*(1, *D*) represents the behavior of staying on the owner’s body for a while.

#### 2.2.4. Communicating Behavior

The communicating behavior of Pyrrhura Molinae parrots can be divided into two types: flying to the flock and without flying to the flock. Considering that both cases have the same probability of happening, these two types of communication behaviors can be represented as follows:(7)$$X_{i}^{t+1}=\left\{\begin{array}{l} 0.2\times rand\times (1-\frac{t}{T})\times \left(X_{i}^{t}-X_{mean}^{t}\right),\;P_{1}\leq 0.5\\ 0.2\times rand\times \exp(-\frac{t}{rand\times T}),\;P_{1}>0.5 \end{array}\right.$$
where *P*_1_ is a random number in the range of [0, 1].

#### 2.2.5. Fear of Strangers’ Behavior

Pyrrhura Molinae parrots also have the behavior trait of being afraid of strangers. They will fly towards the owner and move away from the strangers. This behavior is formulated as follows:(8)$$X_{i}^{t+1}=X_{i}^{t}+O-L$$(9)$$O=rand\times \cos(0.5\pi \times \frac{t}{T})\times \left(X_{best}-X_{i}^{t}\right)$$(10)$$L=\cos(rand\times \pi )\times {\left(\frac{t}{T}\right)}^{\frac{2}{T}}\times \left(X_{i}^{t}-X_{best}\right)$$
where *O* denotes the behavior of flying towards the owner, and *L* denotes the behavior of moving away from the strangers.

### 2.3. Aerial Search Strategy

The aerial search strategy is developed in Arctic Puffin Optimization (APO) [23], which displays a strong ability for global exploration. The mathematical model of it can be shown in the following equations.(11)$$\vec{X_{i}^{t+1}}=\vec{X_{i}^{t}}+(\vec{X_{i}^{t}}-\vec{X_{r}^{t}})\times levy\left(D\right)+R$$(12)$$R=round\left(0.5\times \left(0.05+rand\right)\right)\times randn$$
where *randn* denotes a random value following the standard normal distribution; *r* is a random integer in [1, *N* – 1]. The *round* denotes a function that is used to round values to the nearest whole number.

### 2.4. Fitness–Distance Balance (FDB) Selection

Fitness–distance balance (FDB) selection is a well-known and effective improvement method applied in metaheuristic algorithms, which is proposed by Kahraman et al. in 2020 [24]. FDB selection can improve the search process of the optimization method by balancing the fitness and the distance between the current agent and the best agent. The first step of FDB is to calculate the distance between the candidate solution and the best solution, as shown in Equation (13).(13)$$D_{Pi}=\sqrt{{\left(x_{i,1}-x_{best,1}\right)}^{2}+{\left(x_{i,2}-x_{best,2}\right)}^{2}+\ldots +(x_{i,D}-x_{best,D})}$$
where *D_Pi_* denotes the Euclidean distance between the *i*-th candidate solution and the best solution. In other cases, Manhattan distance and Minkowski distance can also be adopted as the distance metrics. Then the distance vector *D_P_* for the candidate solution can be obtained, which is shown in Equation (14).(14)$$D_{P}\equiv \left[\begin{array}{c} d_{1}\\ \vdots \\ d_{N} \end{array}\right]$$

In the second step, two factors of fitness and distance are comprehensively considered to obtain the score of each individual, as shown in Equation (15).(15)$$S_{P}=\omega \times norm(f)+(1-\omega )\times norm(D_{P})$$
where *w* is a weight coefficient with the value range [0, 1]; *norm* denotes the normalized operator; *f* is the fitness function vector of the population. The score vector *S_P_* is shown in Equation (16).(16)$$S_{P}\equiv \left[\begin{array}{c} s_{1}\\ \vdots \\ s_{N} \end{array}\right]$$

In this paper, a variant of the FDB called RFDB is selected to use the roulette wheel selection [25], which selects an individual according to the score vector *S_P_*. In the RFDB, the scores of all individuals are summed, which is *S_sum_* = *S*_1_ + *S*_2_ + …+ *S_N_*. Then the probability of each individual being selected is the ratio of the individual’s score to the sum of all scores, which is *S_i_*/*S_sum_*. Therefore, the higher the score, the greater the chance of being selected.

## 3. Improved Parrot Optimizer

### 3.1. Motivation

The motivation to improve the PO is to improve the performance and adaptability of the algorithm to meet the increasingly complex requirements of optimization problems. Although the basic PO displays good performance in some optimization problems, it still has limitations when dealing with complicated nonlinear problems, such as the oral English teaching quality evaluation problem. Therefore, this paper introduces efficient strategies and improving techniques to the PO, so that the enhanced PO can flexibly adjust the search strategies during the searching process to improve its search ability and convergence speed, and reduce the possibility of falling into the local optima.

### 3.2. Proposal for IPO

To enhance the global and local search ability of basic PO, several modifications are applied to it, including aerial search strategy, new staying behavior, and roulette fitness–distance balance selection. The details are shown below.

#### 3.2.1. New Exploration Equations Using Aerial Search Strategy

The original PO has a weak exploration phase, so we try to enhance it by using the aerial search strategy that existed in Arctic Puffin Optimization, as shown in Equations (11) and (12).

#### 3.2.2. New Staying Behavior

The staying behavior of PO includes flying to the host and randomly stopping at the host’s body. To increase the local search of the PO, it is assumed that the parrot is already on its owner’s body and might move randomly. Thus, according to Equation (6), the modified staying behavior is described as follows:(17)$$X_{i}^{t+1}=\left\{\begin{array}{l} X_{i}^{t}+X_{best}\times levy\left(D\right)+rand\times ones\left(1,\;D\right),\;P_{2}\leq 0.5\\ X_{i}^{t}+rand\times ones\left(1,\;D\right),\;P_{2}>0.5 \end{array}\right.$$
where *P*_2_ is a random value between 0 and 1, indicating the same possibility of flying and moving for a parrot.

#### 3.2.3. New Communicating Behavior

In the proposed IPO, the RFDB selection is applied to the process of communicating behavior to balance the global and local exploration of PO. The improved communicating behavior can be represented as follows:(18)$$X_{i}^{t+1}=\left\{\begin{array}{l} 0.2\times rand\times (1-\frac{t}{T})\times \left(X_{i}^{t}-X_{mean}^{t}\right),\;P_{3}\leq 0.5\\ X_{RFDB}^{t}+0.2\times rand\times \exp(-\frac{t}{rand\times T}),\;P_{3}>0.5 \end{array}\right.$$
where *P*_3_ is a random value between 0 and 1. $X_{RFDB}^{t}$ is the selected agent using the RFDB selection method.

#### 3.2.4. Architecture of the Proposed IPO

The Pseudocode of IPO is given in Algorithm 1, and the flowchart of IPO is shown in Figure 2. When the IPO begins the optimizing process, the positions of the population are first initialized and the optimal individual is determined. Then, the algorithm enters a cyclic iterative process, and each individual performs a certain behavior according to the parameter *St*, including aerial search strategy, modified staying behavior, improved communicating behavior, and fear of strangers’ behavior. When the number of algorithm iterations reaches the terminating condition, the loop exits and outputs the found optimal solution.
**Algorithm 1:** Pseudocode of the Proposed IPO1. Initialize the IPO parameters: population size *N*, maximum iterations *T*.2. Initialize the population’s positions randomly and identify the best agent.3. **For *t* = 1:*T***4.        Calculate the fitness function.5.        Find the best agent.6.        **For *i* = 1:*N***7.               *St* = randi([1, 4])8.               **If *St* == 1**9.                       Behavior 1: aerial search strategy10.                      Update position by Equations (11) and (12).11.               **Elseif *St* == 2**12.                      Behavior 2: new staying behavior13.                      Update position by Equation (17).14.               **Elseif *St* == 3**15.                      Behavior 3: new communicating behavior16.                      Update position by Equation (18).17.               **Elseif *St* == 4**18.                      Behavior 4: fear of strangers’ behavior19.                      Update position by Equations (8)–(10).20.               **End if**21.               *i* = *i* + 122.        **End for**23.        *t* = *t* + 124. **End For**25. Return the best solution

#### 3.2.5. The Computational Complexity Analysis of IPO

The computational complexity is an important indicator for the performance evaluation of optimization methods [5]. Set the population size to *N*, the maximum number of iterations to *T*, and the dimension of the objective function to *D*. For the original PO, the computational complexity of initialization is *O*(*N* × *D*). The computational complexity during the iterations is *O*(*N* × *D* × *T*). Thus, the total computational complexity of PO is *O*(*N* × *D*) + *O*(*N* × *D* × *T*) = *O*(*N* × *D* × (1 + *T*)). For the IPO, all the applied modifications will not increase the computational complexity. Therefore, the computational complexity of the proposed IPO is the same as the PO, which is *O*(*N* × *D* × (1 + *T*)).

## 4. Experimental Results

In this section, the optimization performance of IPO is evaluated using two types of test experiments, which are CEC2022 benchmark functions [26] and MLP model training. The results of IPO are compared with six other advanced algorithms, including PO [21], Harris Hawks Optimization (HHO) [27], Sine Cosine Algorithm (SCA) [28], Osprey optimization algorithm (OOA) [29], AOA [5], and Aquila Optimizer (AO) [30]. The detailed control parameters of these algorithms are provided in Table 2. To ensure a fair comparison of experimental results, all experiments were independently run 30 times, and for each optimization problem, the maximum number of iterations of all optimization algorithms is set to 500, and the population size is set to 30. The experiments are performed by MATLAB R2021b on a PC with Windows 11 operating system and Intel(R) Core(TM) i7-10700 CPU @ 2.90 GHz and 32.00 GB RAM.

### 4.1. Case 1: CEC2022 Test Sets

The proposed IPO is first tested by using twelve CEC2022 benchmark functions [10,31], where F1 is a unimodal function, F2–F5 are multimodal functions, F6–F8 are hybrid functions, and F9–F12 are composition functions. The details of test functions are presented in Table 3. The range indicates the search range of search space. *F*min indicates the theoretical optimal value. The dimensions of all problems are set to 20.

#### 4.1.1. Ablation Test

This section analyzes the influence of different strategies on the optimization performance of the PO algorithm, including aerial search strategy, new staying behavior, and new communicating behavior. Table 4 shows different versions of the improved PO algorithm, where one denotes that a strategy is adopted, and zero means a strategy is not adopted.

Table 5 provides the results of the sensitivity analysis for the original PO and all other variants from IPO1 to IPO7 algorithms on the CEC2022 test functions. It can be observed that IPO7 has better optimization results than other algorithms. It has ranked the 1st in F1−F3, F5, and F7−F11 whereas it ranked 2nd only in F4 (after IPO4) and F6 (after IPO5). Therefore, the three improvement strategies are helpful to improve the optimization performance of the PO algorithm. From this point forward, we will refer to IPO7 simply as IPO.

#### 4.1.2. Comparison and Analysis with Other Methods

Table 6 shows the test results of mean, best, worst, and standard deviation for all algorithms. The best results of mean values for these test functions are marked in bold. It can be seen from Table 6 that the IPO algorithm outperforms the comparison algorithms in test functions F1, F2, F5, F6, F9, F1, and F12, which shows a strong competitive performance. It is also noted that AO obtains the best mean values on F3, F4, F7, and F8, and PO wins on F10. Overall, the proposed IPO displays obviously better performance than other algorithms.

Moreover, Table 7 gives the results of the Friedman ranking test. It is clearly shown that IPO ranks first compared to the other six algorithms, while AO ranks second and basic PO ranks third. It is further proved that the proposed IPO is superior to other algorithms on CEC2022 test functions.

Table 8 shows the *p*-values results of the Wilcoxon signed-rank test. In this test, The optimization results of IPO are compared respectively with those of each algorithm. The number of the comparison results is 15. *p*-values smaller than 0.05 indicate that the function results of IPO and comparative algorithms have significant differences. Otherwise, there is no significant difference between them. Moreover, the signs ”+/=/–” denote the results of IPO are better, equal, and worse than those of compare algorithms. From Table 8, it is obviously found that IPO has a significant difference compared to the other six methods and shows better optimization performance.

Table 9 reports the results of the average computational time on each test function. It can be seen that due to the addition of several improvement strategies, the calculation time of the IPO is slightly longer than that of the PO. The AOA algorithm has the shortest calculation time because of its simple structure, while the AO algorithm is more complex and has the longest calculation time.

Figure 3 displays the convergence curves of all algorithms on each of the CEC2022 test functions. It can be seen that the IPO converges faster than other algorithms in most cases. In the early stage of the iteration process, IPO can quickly identify better solutions to avoid local optima, such as F2-F6, and F11. Meanwhile, in the subsequent iteration process, the accuracy of the solution can be continuously improved, such as F1 and F6, which proves the effectiveness of the applied improvement strategies. Therefore, IPO shows excellent exploration and exploitation abilities for solving the CEC2022 problems.

Figure 4 presents the boxplots of all algorithms on each CEC2022 test function. It is shown that the boxplots of IPO on F1, F6, F9, F11, and F12 are noticeably narrower than other algorithms, indicating its good stability when solving these problems. In addition, the medians of boxplots for IPO on F1, F5, F9, F11, and F12 are also significantly lower than other methods, showing better accuracy. In other cases, IPO also presents competitive results. Thus, IPO has the merits for solving these optimization problems.

### 4.2. Case 2: Standard Classification Datasets

The second test contains five standard classification datasets, which are XOR, Ballon, Iris, Breast cancer, and Heart [32]. The details of these datasets are provided in Table 10. The MLP models are applied to solve these classification problems. It can be seen that each classification dataset has a corresponding MLP with different structures and parameters.

The training results of IPO-MLP and other compared methods are shown in Table 11. The best mean results are marked in bold. It is found that IPO-MLP has obtained the best mean results among these methods and ranks first at the final rank. In particular, IPO-MLP has achieved significantly lower MSE results in the Balloon dataset. Therefore, IPO-MLP exhibits superior performance in the training of MLP models.

Table 12 gives the results of classification accuracy results. The bold values mean the best mean results. It is observed that although the training results of IPO-MLP are the best, its classification accuracy is not the highest on Breast cancer and Heart datasets. Overall, IPO-MLP has obtained competitive results on these five classification datasets. The Friedman ranking results show that the IPO-MLP is the best.

### 4.3. Case 3: Oral English Education Evaluation Problem

In the third case, the proposed IPO is used to solve the oral English education evaluation problem. The oral English education evaluation problem can be regarded as a classification problem with multiple features. In this paper, the evaluation model is constructed using a multilayer perceptron model with a 10-21-3 structure. Then, the weights and biases of this MLP model are optimized by using the proposed IPO. The results are shown below.

#### 4.3.1. Indexes of Oral English Teaching Quality Evaluation Model

For better evaluating the oral English teaching quality, the indexes of oral English teaching quality evaluation problem are selected according to the works in [19]. The index system is constructed as shown in Figure 5, which has five first-grade indexes: qualities of teacher, teaching attitude, teaching content, teaching method, and teaching effectiveness. In each first-grade index, there are one or more second-grade indexes. These elements play a leading role in the oral English teaching evaluation and ensure the scientific and reasonable teaching quality evaluation system.

As can be seen in Figure 5, ten features in the oral English education evaluation problem are selected to determine the evaluation outcomes of teachers. The evaluation results are divided into three cases, which are excellent, good, and qualified. Therefore, an MLP model with a 10-21-3 structure is constructed to find the relationship between the indexes and evaluation outcomes. The node number of the hidden layer is determined by an empirical formula 2 × *n* + 1 [33], where *n* is the number of input parameters.

#### 4.3.2. Analysis of Testing Results

For the experiments, a total of 60 groups of oral English quality evaluation data were collected, of which 20 are excellent, 20 are good, and 20 are qualified. The proposed IPO and other six compared methods are employed to train the constructed MLP models. The experiments are conducted for 30 times. The results are shown in Figure 6, Table 13, and Table 14.

Figure 6 shows the convergence curves of average MSE during the training process. It is observed that the IPO presents the ability to consistently find smaller MSE values throughout the whole process and obtain the smallest value at last, while the original PO shows poor search ability for this problem. It is also noted that HHO and AO also show good optimization results compared to IPO. Table 13 provides the training results of all algorithms. It can be found that IPO has the best performance for training the evaluation model. In terms of the mean value index, IPO-MLP obtains the lowest value with 4.111E-02 and ranks first among all algorithms.

Meanwhile, the accuracy results are given in Table 14. It can be seen that IPO-MLP also ranks first with a mean accuracy of 7.022 × 10^1^. The IPO-MLP also obtains the highest accuracy of 8.833 × 10^1^. Therefore, the proposed IPO has the merits of solving the oral English teaching quality evaluation problem. By using the proposed IPO-MLP, the decision-makers can better allocate teaching resources based on the accurate assessment of teachers’ teaching levels.

## 5. Conclusions and Future Work

In this paper, the aerial search strategy, modified staying behavior, and roulette fitness–distance balance selection are used to improve the basic PO for better optimization performance. The proposed IPO was tested on twelve CEC2022 test functions and applied to optimize the parameters of MLP models for classification problems. The results of the IPO significantly outperformed the other six advanced algorithms, including PO, HHO, SCA, OOA, AOA, and AO. An evaluation model of oral English teaching quality is also constructed using an MLP model with a 10-21-3 structure. The IPO is used to optimize the weight and bias parameters of the evaluation model. The results show that the IPO-MLP model can more accurately evaluate the outcomes of oral English teaching quality and has obtained the highest accuracy of 88.33%.

In future work, the suggested optimizer can be extended to tackle a broader range of complex optimization problems, such as robotic path planning, feature selection in high-dimensional datasets, and dynamic scheduling tasks. Additionally, the IPO-MLP framework can be integrated with other metaheuristic algorithms to enhance convergence speed and solution quality. Another promising direction is the automatic optimization of MLP architecture, including activation function selection, hidden layer configuration, and neuron count, potentially using self-adaptive or co-evolutionary strategies. These enhancements can further improve the robustness and generalization ability of the proposed approach across diverse application domains.

## Figures and Tables

**Figure 1 biomimetics-10-00358-f001:**
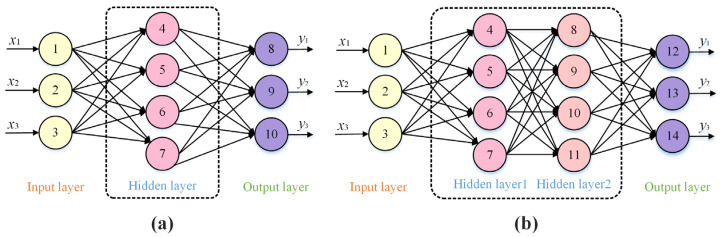
Structure of MLP model: (**a**) MLP with 3-4-3 structure. (**b**) MLP with 3-4-4-3 structure.

**Figure 2 biomimetics-10-00358-f002:**
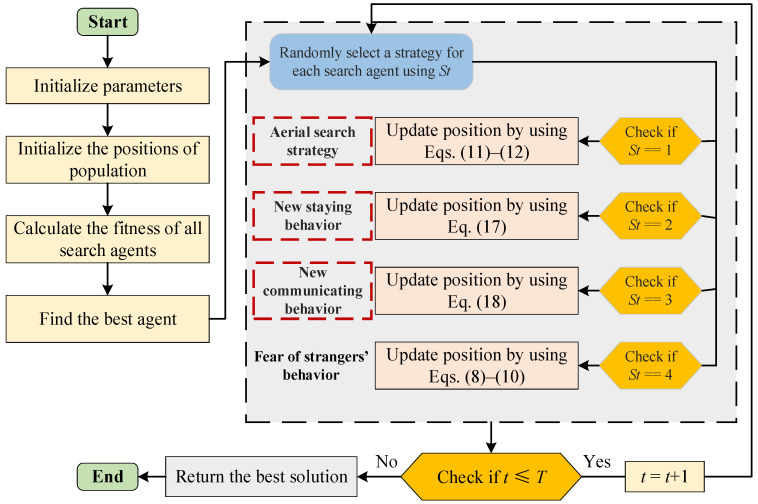
The flowchart of IPO.

**Figure 3 biomimetics-10-00358-f003:**
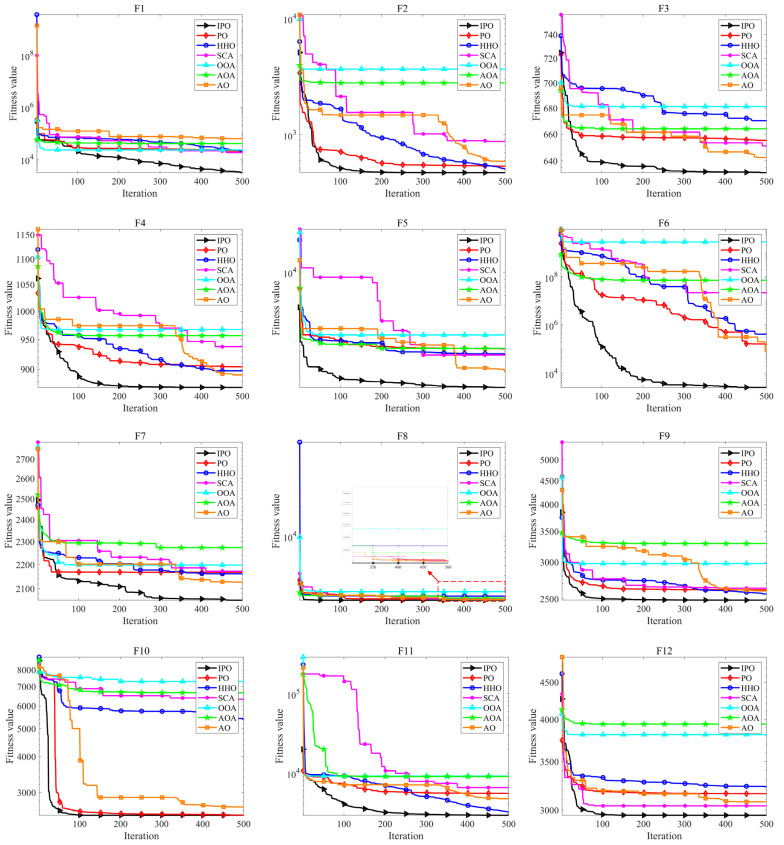
Convergence curves of all algorithms on CEC2022 test functions.

**Figure 4 biomimetics-10-00358-f004:**
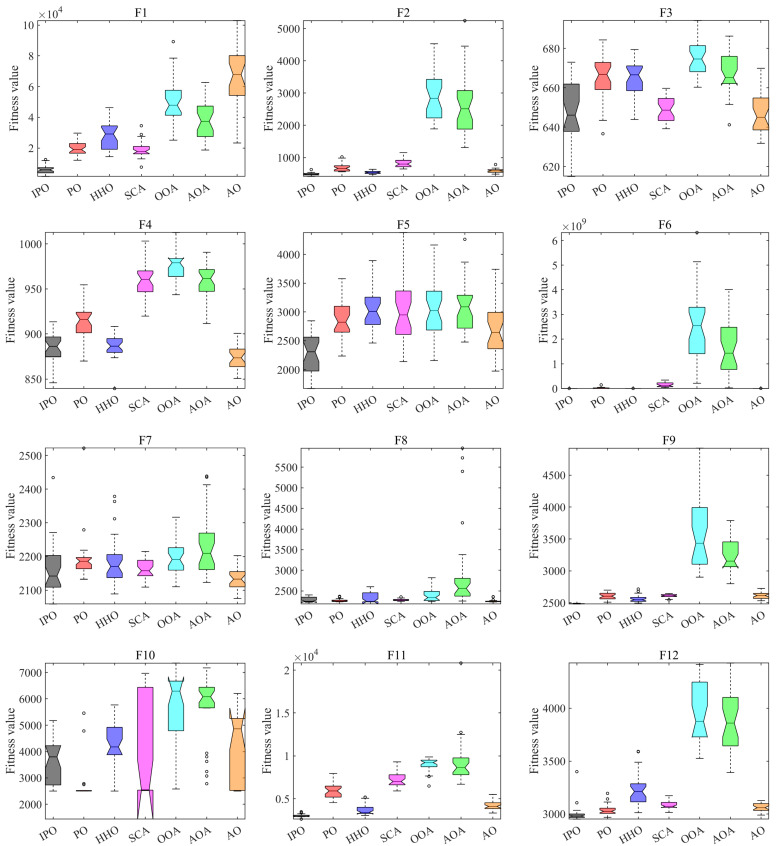
Boxplots of all algorithms on CEC2022 test functions.

**Figure 5 biomimetics-10-00358-f005:**
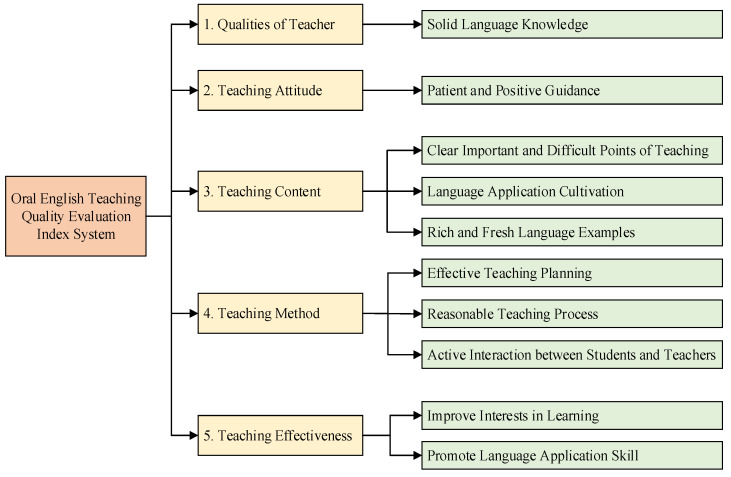
Oral English teaching quality evaluation index system.

**Figure 6 biomimetics-10-00358-f006:**
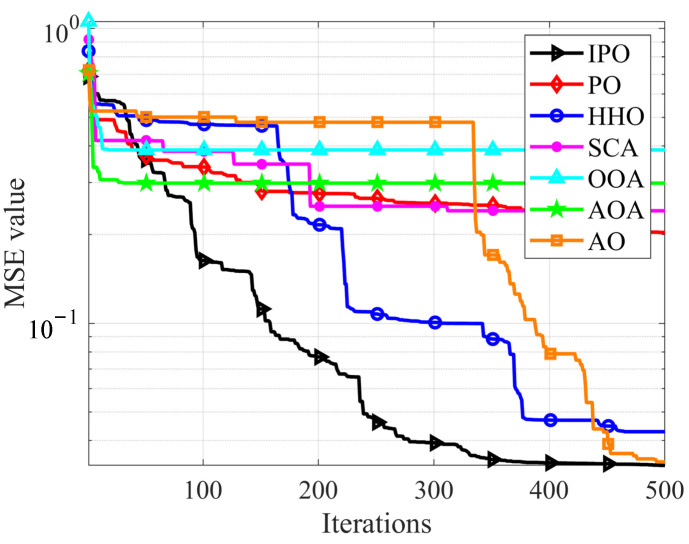
The convergence curves of average MSE for all algorithms.

**Table 1 biomimetics-10-00358-t001:** Overview of recent literature studies on English teaching quality evaluation.

Author	Year	Algorithm	Model	Remarks
Zhang [13]	2018	Principal component analysis	Support vector machine	English teaching quality evaluation
Lu [16]	2021	Genetic algorithm	RBF neural network	Evaluation of English interpretation teaching quality
Wei [17]	2022	Improved quantum particle swarm algorithm	Support vector machine	Classification of college English teaching effects
Zhang [18]	2022	Particle swarm algorithm	Least squares support vector machine	Evaluation of College English Teaching Effect
Tan [19]	2023	Improved crow search algorithm	BP neural network	Evaluation of oral English teaching quality
Miao [20]	2023	Decision tree algorithm	-	Evaluation of the teaching effect of oral English teaching

**Table 2 biomimetics-10-00358-t002:** The parameter settings of the proposed algorithm and the compared algorithms.

Algorithms	Year	Parameters
IPO	-	*P* ∈ [0, 1]
PO [21]	2024	*P* ∈ [0, 1]
HHO [27]	2019	*β* = 1.5; *E*_0_ ∈ [−1, 1]
SCA [28]	2016	*a* is linearly decreased from 2 to 0
OOA [29]	2023	*r*_1_ ∈ [0, 1]
AOA [5]	2021	*α* = 5; *µ* = 0.499
AO [30]	2021	*α* = 0.1; *δ* = 0.1

**Table 3 biomimetics-10-00358-t003:** The detailed information on the CEC2022 test functions.

Function Type	No.	Description	Range	*F*min
Unimodal function	F1	Shifted and full Rotated Zakharov Function	[−100, 100]	300
Simple multimodal functions	F2	Shifted and full Rotated Rosenbrock’s Function	[−100, 100]	400
	F3	Shifted and full Rotated Expanded Schaffer’s *f*6 Function	[−100, 100]	600
	F4	Shifted and full Rotated Non-Continuous Rastrigin’s Function	[−100, 100]	800
	F5	Shifted and full Rotated Levy Function	[−100, 100]	900
Hybrid functions	F6	Hybrid Function 1 (*N* = 3)	[−100, 100]	1800
	F7	Hybrid Function 2 (*N* =6)	[−100, 100]	2000
	F8	Hybrid Function 3 (*N* = 5)	[−100, 100]	2200
Composition functions	F9	Composition Function 1 (*N* = 5)	[−100, 100]	2300
	F10	Composition Function 2 (*N* = 4)	[−100, 100]	2400
	F11	Composition Function 3 (*N* = 5)	[−100, 100]	2600
	F12	Composition Function 4 (*N* = 6)	[−100, 100]	2700

**Table 4 biomimetics-10-00358-t004:** PO and improved PO with different strategies.

Algorithms	Aerial Search	New Staying Behavior	New Communicating Behavior
PO	0	0	0
IPO1	1	0	0
IPO2	0	1	0
IPO3	0	0	1
IPO4	1	1	0
IPO5	1	0	1
IPO6	0	1	1
IPO7	1	1	1

**Table 5 biomimetics-10-00358-t005:** Sensitivity analysis of PO and its improved versions.

Function	PO	IPO1	IPO2	IPO3	IPO4	IPO5	IPO6	IPO7
F1	1.205 × 10^4^	7.841 × 10^3^	9.183 × 10^3^	7.080 × 10^3^	3.926 × 10^3^	2.296 × 10^3^	8.056 × 10^3^	**1.525 × 10^3^**
F2	5.733 × 10^2^	4.935 × 10^2^	5.793 × 10^2^	5.327 × 10^2^	4.616 × 10^2^	4.542 × 10^2^	4.925 × 10^2^	**4.151 × 10^2^**
F3	6.366 × 10^2^	6.322 × 10^2^	6.394 × 10^2^	6.378 × 10^2^	6.230 × 10^2^	6.274 × 10^2^	6.365 × 10^2^	**6.150 × 10^2^**
F4	8.700 × 10^2^	8.813 × 10^2^	9.006 × 10^2^	8.761 × 10^2^	8.389 × 10^2^	8.703 × 10^2^	8.912 × 10^2^	8.458 × 10^2^
F5	2.252 × 10^3^	2.060 × 10^3^	2.186 × 10^3^	1.936 × 10^3^	1.832 × 10^3^	1.885 × 10^3^	1.779 × 10^3^	**1.671 × 10^3^**
F6	3.097 × 10^4^	4.143 × 10^3^	1.507 × 10^5^	7.059 × 10^4^	2.472 × 10^3^	2.160 × 10^3^	2.245 × 10^4^	2.215 × 10^3^
F7	2.132 × 10^3^	2.112 × 10^3^	2.087 × 10^3^	2.093 × 10^3^	2.117 × 10^3^	2.067 × 10^3^	2.118 × 10^3^	**2.059 × 10^3^**
F8	2.237 × 10^3^	2.228 × 10^3^	2.232 × 10^3^	2.231 × 10^3^	2.230 × 10^3^	2.230 × 10^3^	2.235 × 10^3^	**2.227 × 10^3^**
F9	2.514 × 10^3^	2.483 × 10^3^	2.528 × 10^3^	2.498 × 10^3^	2.481 × 10^3^	2.482 × 10^3^	2.496 × 10^3^	**2.481 × 10^3^**
F10	2.501 × 10^3^	2.501 × 10^3^	2.501 × 10^3^	2.501 × 10^3^	2.501 × 10^3^	2.501 × 10^3^	2.501 × 10^3^	**2.501 × 10^3^**
F11	4.552 × 10^3^	3.763 × 10^3^	4.300 × 10^3^	4.184 × 10^3^	2.850 × 10^3^	2.786 × 10^3^	4.665 × 10^3^	**2.625 × 10^3^**
F12	2.968 × 10^3^	2.942 × 10^3^	2.963 × 10^3^	2.954 × 10^3^	2.942 × 10^3^	2.946 × 10^3^	2.973 × 10^3^	2.950 × 10^3^

The bold values indicate the best values.

**Table 6 biomimetics-10-00358-t006:** Numerical results of IPO and other compared algorithms on CEC2022 test functions.

Function	Item	IPO	PO	HHO	SCA	OOA	AOA	AO
F1	Mean	**6.184 × 10^3^**	1.949 × 10^4^	2.778 × 10^4^	1.886 × 10^4^	5.002 × 10^4^	3.783 × 10^4^	6.688 × 10^4^
	Best	1.525 × 10^3^	1.205 × 10^4^	1.441 × 10^4^	7.572 × 10^3^	2.513 × 10^4^	1.870 × 10^4^	2.326 × 10^4^
	Worst	1.249 × 10^4^	2.970 × 10^4^	4.631 × 10^4^	3.451 × 10^4^	8.921 × 10^4^	6.266 × 10^4^	1.030 × 10^5^
	Std	3.255 × 10^3^	4.859 × 10^3^	8.889 × 10^3^	5.317 × 10^3^	1.523 × 10^4^	1.243 × 10^4^	1.920 × 10^4^
F2	Mean	**4.843 × 10^2^**	6.911 × 10^2^	5.458 × 10^2^	8.207 × 10^2^	2.946 × 10^3^	2.587 × 10^3^	5.909 × 10^2^
	Best	4.151 × 10^2^	5.586 × 10^2^	4.758 × 10^2^	6.461 × 10^2^	1.889 × 10^3^	1.312 × 10^3^	4.872 × 10^2^
	Worst	6.280 × 10^2^	1.020 × 10^3^	6.360 × 10^2^	1.154 × 10^3^	4.530 × 10^3^	5.251 × 10^3^	7.844 × 10^2^
	Std	4.196 × 10^1^	1.206 × 10^2^	4.525 × 10^1^	1.260 × 10^2^	7.930 × 10^2^	9.187 × 10^2^	6.130 × 10^1^
F3	Mean	6.468 × 10^2^	6.650 × 10^2^	6.654 × 10^2^	6.494 × 10^2^	6.752 × 10^2^	6.669 × 10^2^	**6.466 × 10^2^**
	Best	6.150 × 10^2^	6.366 × 10^2^	6.439 × 10^2^	6.392 × 10^2^	6.603 × 10^2^	6.412 × 10^2^	6.318 × 10^2^
	Worst	6.730 × 10^2^	6.843 × 10^2^	6.794 × 10^2^	6.597 × 10^2^	6.942 × 10^2^	6.863 × 10^2^	6.699 × 10^2^
	Std	1.497 × 10^1^	1.160 × 10^1^	8.112 × 10^0^	6.388 × 10^0^	8.921 × 10^0^	1.025 × 10^1^	9.346 × 10^0^
F4	Mean	8.849 × 10^2^	9.144 × 10^2^	8.862 × 10^2^	9.590 × 10^2^	9.741 × 10^2^	9.586 × 10^2^	**8.747 × 10^2^**
	Best	8.458 × 10^2^	8.700 × 10^2^	8.394 × 10^2^	9.198 × 10^2^	9.435 × 10^2^	9.115 × 10^2^	8.507 × 10^2^
	Worst	9.134 × 10^2^	9.546 × 10^2^	9.083 × 10^2^	1.003 × 10^3^	1.013 × 10^3^	9.907 × 10^2^	9.006 × 10^2^
	Std	1.538 × 10^1^	1.813 × 10^1^	1.312 × 10^1^	1.818 × 10^1^	1.698 × 10^1^	1.945 × 10^1^	1.273 × 10^1^
F5	Mean	**2.308 × 10^3^**	2.856 × 10^3^	3.037 × 10^3^	3.009 × 10^3^	3.057 × 10^3^	3.083 × 10^3^	2.699 × 10^3^
	Best	1.671 × 10^3^	2.234 × 10^3^	2.460 × 10^3^	2.140 × 10^3^	2.158 × 10^3^	2.477 × 10^3^	1.975 × 10^3^
	Worst	2.845 × 10^3^	3.577 × 10^3^	3.892 × 10^3^	4.381 × 10^3^	4.163 × 10^3^	4.262 × 10^3^	3.742 × 10^3^
	Std	3.376 × 10^2^	3.425 × 10^2^	3.727 × 10^2^	5.836 × 10^2^	5.307 × 10^2^	3.962 × 10^2^	4.439 × 10^2^
F6	Mean	**6.301 × 10^3^**	1.573 × 10^7^	2.003 × 10^5^	1.565 × 10^8^	2.642 × 10^9^	1.622 × 10^9^	6.389 × 10^5^
	Best	2.116 × 10^3^	2.660 × 10^4^	7.384 × 10^4^	4.061 × 10^7^	2.125 × 10^8^	2.472 × 10^7^	5.628 × 10^4^
	Worst	1.997 × 10^4^	1.467 × 10^8^	6.035 × 10^5^	3.427 × 10^8^	6.325 × 10^9^	4.011 × 10^9^	4.598 × 10^6^
	Std	4.489 × 10^3^	2.812 × 10^7^	1.056 × 10^5^	9.833 × 10^7^	1.415 × 10^9^	1.106 × 10^9^	8.337 × 10^5^
F7	Mean	2.160 × 10^3^	2.194 × 10^3^	2.188 × 10^3^	2.163 × 10^3^	2.198 × 10^3^	2.237 × 10^3^	**2.133 × 10^3^**
	Best	2.059 × 10^3^	2.132 × 10^3^	2.089 × 10^3^	2.109 × 10^3^	2.110 × 10^3^	2.123 × 10^3^	2.075 × 10^3^
	Worst	2.434 × 10^3^	2.522 × 10^3^	2.379 × 10^3^	2.215 × 10^3^	2.317 × 10^3^	2.439 × 10^3^	2.203 × 10^3^
	Std	7.438 × 10^1^	6.810 × 10^1^	7.016 × 10^1^	2.834 × 10^1^	4.934 × 10^1^	9.596 × 10^1^	2.950 × 10^1^
F8	Mean	2.278 × 10^3^	2.273 × 10^3^	2.319 × 10^3^	2.283 × 10^3^	2.396 × 10^3^	2.924 × 10^3^	**2.256 × 10^3^**
	Best	2.226 × 10^3^	2.236 × 10^3^	2.235 × 10^3^	2.248 × 10^3^	2.241 × 10^3^	2.255 × 10^3^	2.232 × 10^3^
	Worst	2.403 × 10^3^	2.368 × 10^3^	2.601 × 10^3^	2.354 × 10^3^	2.820 × 10^3^	5.963 × 10^3^	2.359 × 10^3^
	Std	5.843 × 10^1^	4.383 × 10^1^	1.157 × 10^2^	2.381 × 10^1^	1.622 × 10^2^	1.017 × 10^3^	3.545 × 10^1^
F9	Mean	**2.485 × 10^3^**	2.608 × 10^3^	2.558 × 10^3^	2.609 × 10^3^	3.541 × 10^3^	3.242 × 10^3^	2.616 × 10^3^
	Best	2.481 × 10^3^	2.510 × 10^3^	2.496 × 10^3^	2.546 × 10^3^	2.902 × 10^3^	2.799 × 10^3^	2.532 × 10^3^
	Worst	2.493 × 10^3^	2.698 × 10^3^	2.715 × 10^3^	2.645 × 10^3^	4.923 × 10^3^	3.789 × 10^3^	2.725 × 10^3^
	Std	3.557 × 10^0^	5.436 × 10^1^	5.194 × 10^1^	2.606 × 10^1^	5.358 × 10^2^	2.493 × 10^2^	5.203 × 10^1^
F10	Mean	3.680 × 10^3^	**2.700 × 10^3^**	4.171 × 10^3^	4.028 × 10^3^	5.627 × 10^3^	5.670 × 10^3^	4.180 × 10^3^
	Best	2.501 × 10^3^	2.501 × 10^3^	2.501 × 10^3^	2.519 × 10^3^	2.579 × 10^3^	2.780 × 10^3^	2.501 × 10^3^
	Worst	5.181 × 10^3^	5.457 × 10^3^	5.767 × 10^3^	6.969 × 10^3^	7.359 × 10^3^	7.175 × 10^3^	6.202 × 10^3^
	Std	8.421 × 10^2^	6.670 × 10^2^	9.453 × 10^2^	1.953 × 10^3^	1.460 × 10^3^	1.218 × 10^3^	1.283 × 10^3^
F11	Mean	**3.001 × 10^3^**	5.945 × 10^3^	3.684 × 10^3^	7.217 × 10^3^	8.982 × 10^3^	9.271 × 10^3^	4.261 × 10^3^
	Best	2.624 × 10^3^	4.552 × 10^3^	3.057 × 10^3^	5.914 × 10^3^	6.490 × 10^3^	6.698 × 10^3^	3.336 × 10^3^
	Worst	3.481 × 10^3^	7.952 × 10^3^	5.192 × 10^3^	9.309 × 10^3^	9.882 × 10^3^	2.082 × 10^4^	5.502 × 10^3^
	Std	2.001 × 10^2^	8.775 × 10^2^	6.229 × 10^2^	8.595 × 10^2^	7.743 × 10^2^	2.739 × 10^3^	6.180 × 10^2^
F12	Mean	**2.999 × 10^3^**	3.036 × 10^3^	3.229 × 10^3^	3.085 × 10^3^	3.936 × 10^3^	3.886 × 10^3^	3.061 × 10^3^
	Best	2.950 × 10^3^	2.968 × 10^3^	3.011 × 10^3^	3.014 × 10^3^	3.526 × 10^3^	3.392 × 10^3^	2.989 × 10^3^
	Worst	3.400 × 10^3^	3.195 × 10^3^	3.592 × 10^3^	3.172 × 10^3^	4.416 × 10^3^	4.429 × 10^3^	3.126 × 10^3^
	Std	8.243 × 10^1^	4.866 × 10^1^	1.475 × 10^2^	4.117 × 10^1^	2.725 × 10^2^	2.795 × 10^2^	3.982 × 10^1^

The bold values indicate the best values.

**Table 7 biomimetics-10-00358-t007:** Friedman ranking results of all algorithms on CEC2022 test functions.

Function	IPO	PO	HHO	SCA	OOA	AOA	AO
F1	1	3	4	2	6	5	7
F2	1	4	2	5	7	6	3
F3	2	4	5	3	7	6	1
F4	2	4	3	6	7	5	1
F5	1	3	5	4	6	7	2
F6	1	4	2	5	7	6	3
F7	2	5	4	3	6	7	1
F8	3	2	5	4	6	7	1
F9	1	3	2	4	7	6	5
F10	2	1	4	3	6	7	5
F11	1	4	2	5	6	7	3
F12	1	2	5	4	7	6	3
Mean Rank	1.50	3.25	3.58	4.00	6.50	6.25	2.92
Final Rank	1	3	4	5	7	6	2

**Table 8 biomimetics-10-00358-t008:** Wilcoxon signed-rank test *p*-values results of IPO compared to other algorithms on CEC2022 test functions.

Function	vs. PO	vs. HHO	vs. SCA	vs. OOA	vs. AOA	vs. AO
F1	4.143 × 10^−6^	3.392 × 10^−6^	6.152 × 10^−6^	3.392 × 10^−6^	3.392 × 10^−6^	3.392 × 10^−6^
F2	9.073 × 10^−6^	1.140 × 10^−2^	3.392 × 10^−6^	3.392 × 10^−6^	3.392 × 10^−6^	1.146 × 10^−4^
F3	4.937 × 10^−4^	1.050 × 10^−3^	4.553 × 10^−1^	1.330 × 10^−5^	1.866 × 10^−3^	2.455 × 10^−1^
F4	4.020 × 10^−5^	8.357 × 10^−1^	3.392 × 10^−6^	3.392 × 10^−6^	3.392 × 10^−6^	1.057 × 10^−1^
F5	1.866 × 10^−3^	4.020 × 10^−5^	3.691 × 10^−3^	2.229 × 10^−4^	1.146 × 10^−4^	2.463 × 10^−3^
F6	3.392 × 10^−6^	3.392 × 10^−6^	3.392 × 10^−6^	3.392 × 10^−6^	3.392 × 10^−6^	3.392 × 10^−6^
F7	6.783 × 10^−1^	1.844 × 10^−1^	3.195 × 10^−1^	6.187 × 10^−1^	2.998 × 10^−1^	3.440 × 10^−2^
F8	6.783 × 10^−1^	9.669 × 10^−1^	6.783 × 10^−1^	5.452 × 10^−3^	4.937 × 10^−4^	7.089 × 10^−1^
F9	3.392 × 10^−6^	3.392 × 10^−6^	3.392 × 10^−6^	3.392 × 10^−6^	3.392 × 10^−6^	3.392 × 10^−6^
F10	1.440 × 10^−2^	3.837 × 10^−1^	9.339 × 10^−1^	6.709 × 10^−4^	1.892 × 10^−4^	1.440 × 10^−2^
F11	3.392 × 10^−6^	2.798 × 10^−5^	3.392 × 10^−6^	3.392 × 10^−6^	3.392 × 10^−6^	3.392 × 10^−6^
F12	1.807 × 10^−2^	1.330 × 10^−5^	4.020 × 10^−5^	3.392 × 10^−6^	3.392 × 10^−6^	9.059 × 10^−4^
+/=/–	9/2/1	8/4/0	8/4/0	8/4/0	8/4/0	8/3/1

**Table 9 biomimetics-10-00358-t009:** The average computational time for each algorithm on CEC2022 test functions.

Function	IPO	PO	HHO	SCA	OOA	AOA	AO
F1	1.589 × 10^−1^	1.311 × 10^−1^	1.848 × 10^−1^	1.219 × 10^−1^	1.359 × 10^−1^	1.165 × 10^−1^	2.057 × 10^−1^
F2	1.562 × 10^−1^	1.346 × 10^−1^	1.786 × 10^−1^	1.280 × 10^−1^	1.404 × 10^−1^	1.191 × 10^−1^	2.017 × 10^−1^
F3	2.174 × 10^−1^	1.930 × 10^−1^	3.393 × 10^−1^	1.834 × 10^−1^	2.550 × 10^−1^	1.691 × 10^−1^	3.094 × 10^−1^
F4	1.687 × 10^−1^	1.532 × 10^−1^	2.316 × 10^−1^	1.425 × 10^−1^	1.649 × 10^−1^	1.261 × 10^−1^	2.305 × 10^−1^
F5	1.700 × 10^−1^	1.464 × 10^−1^	2.399 × 10^−1^	1.414 × 10^−1^	1.706 × 10^−1^	1.297 × 10^−1^	2.329 × 10^−1^
F6	1.560 × 10^−1^	1.373 × 10^−1^	2.034 × 10^−1^	1.316 × 10^−1^	1.457 × 10^−1^	1.193 × 10^−1^	2.043 × 10^−1^
F7	2.315 × 10^−1^	2.135 × 10^−1^	4.130 × 10^−1^	2.111 × 10^−1^	2.955 × 10^−1^	1.902 × 10^−1^	3.616 × 10^−1^
F8	2.459 × 10^−1^	2.293 × 10^−1^	4.406 × 10^−1^	2.347 × 10^−1^	3.348 × 10^−1^	2.095 × 10^−1^	4.006 × 10^−1^
F9	1.856 × 10^−1^	1.728 × 10^−1^	3.057 × 10^−1^	1.685 × 10^−1^	2.407 × 10^−1^	1.569 × 10^−1^	2.968 × 10^−1^
F10	1.638 × 10^−1^	1.548 × 10^−1^	2.709 × 10^−1^	1.565 × 10^−1^	2.081 × 10^−1^	1.381 × 10^−1^	2.549 × 10^−1^
F11	1.927 × 10^−1^	1.804 × 10^−1^	3.216 × 10^−1^	1.840 × 10^−1^	2.652 × 10^−1^	1.733 × 10^−1^	3.218 × 10^−1^
F12	2.139 × 10^−1^	1.980 × 10^−1^	3.667 × 10^−1^	2.016 × 10^−1^	3.029 × 10^−1^	1.857 × 10^−1^	3.558 × 10^−1^

**Table 10 biomimetics-10-00358-t010:** The detailed information of classification datasets.

Datasets	Number of Features	Number of Training Samples	Number of Test Samples	Number of Classes	MLP Structure	Dimension	Search Range
XOR	3	8	8	2	3-7-1	36	[−10, 10]
Balloon	4	20	20	2	4-9-1	55	[−10, 10]
Iris	4	150	150	3	4-9-3	75	[−10, 10]
Breast cancer	9	599	100	2	9-19-1	210	[−10, 10]
Heart	22	80	187	2	22-45-1	1081	[−10, 10]

**Table 11 biomimetics-10-00358-t011:** MSE results on training data.

Datasets	Index	IPO-MLP	PO-MLP	HHO-MLP	SCA-MLP	OOA-MLP	AOA-MLP	AO-MLP
XOR	Mean	**1.186 × 10** ** ^−^ ** ** ^2^ **	1.508 × 10^−1^	3.865 × 10^−2^	4.753 × 10^−2^	1.942 × 10^−1^	2.098 × 10^−1^	4.933 × 10^−2^
	Best	2.274 × 10^−9^	4.259 × 10^−8^	5.663 × 10^−8^	3.263 × 10^−3^	1.216 × 10^−1^	1.367 × 10^−1^	9.035 × 10^−9^
	Worst	2.500 × 10^−1^	2.500 × 10^−1^	2.143 × 10^−1^	1.093 × 10^−1^	2.500 × 10^−1^	2.500 × 10^−1^	2.500 × 10^−1^
	Std	4.887 × 10^−2^	1.029 × 10^−1^	6.754 × 10^−2^	3.419 × 10^−2^	3.971 × 10^−2^	3.530 × 10^−2^	7.784 × 10^−2^
	Rank	1	5	2	3	6	7	4
Balloon	Mean	**3.939 × 10^−11^**	3.739 × 10^−6^	3.058 × 10^−6^	1.095 × 10^−5^	5.539 × 10^−2^	5.416 × 10^−3^	8.642 × 10^−7^
	Best	2.280 × 10^−22^	1.397 × 10^−15^	1.568 × 10^−19^	6.161 × 10^−9^	3.989 × 10^−4^	5.130 × 10^−9^	1.634 × 10^−18^
	Worst	1.181 × 10^−9^	4.288 × 10^−5^	6.335 × 10^−5^	1.219 × 10^−4^	1.524 × 10^−1^	5.789 × 10^−2^	1.385 × 10^−5^
	Std	2.157 × 10^−1^	9.062 × 10^−6^	1.250 × 10^−5^	2.341 × 10^−5^	3.975 × 10^−2^	1.217 × 10^−2^	3.286 × 10^−6^
	Rank	1	4	3	5	7	6	2
Iris	Mean	**3.541 × 10** ** ^−^ ** ** ^2^ **	1.013 × 10^−1^	7.539 × 10^−2^	1.965 × 10^−1^	4.756 × 10^−1^	3.509 × 10^−1^	5.743 × 10^−2^
	Best	1.549 × 10^−2^	4.374 × 10^−2^	3.090 × 10^−2^	7.834 × 10^−2^	1.853 × 10^−1^	2.276 × 10^−1^	2.375 × 10^−2^
	Worst	1.060 × 10^−1^	2.896 × 10^−1^	3.745 × 10^−1^	3.427 × 10^−1^	6.657 × 10^−1^	4.835 × 10^−1^	3.511 × 10^−1^
	Std	1.871 × 10^−2^	5.048 × 10^−2^	8.281 × 10^−2^	6.464 × 10^−2^	1.099 × 10^−1^	7.280 × 10^−2^	8.078 × 10^−2^
	Rank	1	4	3	5	7	6	2
Breast cancer	Mean	**1.499 × 10** ** ^−^ ** ** ^3^ **	1.729 × 10^−3^	1.887 × 10^−3^	1.388 × 10^−2^	2.078 × 10^−3^	4.708 × 10^−3^	1.985 × 10^−3^
	Best	1.329 × 10^−3^	1.486 × 10^−3^	1.618 × 10^−3^	4.188 × 10^−3^	1.698 × 10^−3^	2.256 × 10^−3^	1.639 × 10^−3^
	Worst	1.711 × 10^−3^	2.270 × 10^−3^	2.131 × 10^−3^	3.623 × 10^−2^	2.941 × 10^−3^	1.247 × 10^−2^	2.628 × 10^−3^
	Std	8.498 × 10^−5^	1.775 × 10^−4^	1.520 × 10^−4^	7.986 × 10^−3^	2.530 × 10^−4^	2.191 × 10^−3^	2.365 × 10^−4^
	Rank	1	2	3	7	5	6	4
Heart	Mean	**8.432 × 10** ** ^−^ ** ** ^2^ **	1.134 × 10^−1^	1.226 × 10^−1^	1.810 × 10^−1^	1.689 × 10^−1^	1.533 × 10^−1^	1.008 × 10^−1^
	Best	6.383 × 10^−2^	8.461 × 10^−2^	8.822 × 10^−2^	1.454 × 10^−1^	1.493 × 10^−1^	1.215 × 10^−1^	6.721 × 10^−2^
	Worst	1.192 × 10^−1^	1.450 × 10^−1^	1.699 × 10^−1^	2.113 × 10^−1^	1.937 × 10^−1^	1.798 × 10^−1^	1.410 × 10^−1^
	Std	1.233 × 10^−2^	1.592 × 10^−2^	2.094 × 10^−2^	1.678 × 10^−2^	1.051 × 10^−2^	1.436 × 10^−2^	2.134 × 10^−2^
	Rank	1	3	4	7	6	5	2
Mean Rank		1	3.6	3	5.4	6.2	6	2.8
Final Rank		1	4	3	5	7	6	2

The bold values indicate the best values.

**Table 12 biomimetics-10-00358-t012:** Classification accuracy results on test data (%).

Datasets	Index	IPO-MLP	PO-MLP	HHO-MLP	SCA-MLP	OOA-MLP	AOA-MLP	AO-MLP
XOR	Mean	**94.17**	29.58	66.67	46.67	7.08	9.58	71.67
	Best	100.00	100.00	100.00	75.00	37.50	50.00	100.00
	Worst	0.00	0.00	0.00	12.50	0.00	0.00	0.00
	Std	22.44	40.13	36.31	17.66	10.73	13.80	33.95
	Rank	1	5	3	4	7	6	2
Balloon	Mean	**100.00**	100.00	100.00	100.00	30.67	75.00	100.00
	Best	100.00	100.00	100.00	100.00	100.00	100.00	100.00
	Worst	100.00	100.00	100.00	100.00	0.00	0.00	100.00
	Std	0.00	0.00	0.00	0.00	27.94	34.22	0.00
	Rank	1	1	1	1	7	6	1
Iris	Mean	**75.42**	38.58	48.24	36.18	1.80	2.04	74.93
	Best	92.00	79.33	86.67	64.67	33.33	24.67	89.33
	Worst	14.67	0.00	0.00	9.33	0.00	0.00	41.33
	Std	16.24	24.02	22.44	15.63	6.32	6.21	11.88
	Rank	1	4	3	5	7	6	2
Breast cancer	Mean	98.57	98.53	98.83	58.17	**99.13**	90.00	97.97
	Best	100.00	100.00	100.00	95.00	100.00	99.00	100.00
	Worst	98.00	94.00	98.00	5.00	96.00	0.00	96.00
	Std	0.63	1.25	0.70	33.13	0.78	18.23	0.89
	Rank	4	5	3	7	1	6	2
Heart	Mean	66.96	48.25	40.88	**73.46**	24.96	38.58	62.63
	Best	87.50	76.25	71.25	80.00	35.00	76.25	90.00
	Worst	38.75	3.75	5.00	65.00	0.00	5.00	32.50
	Std	12.05	21.49	16.17	3.39	8.97	19.66	14.28
	Rank	2	4	5	1	7	6	3
Mean Rank		**1.8**	3.8	3	3.6	5.8	6	2
Final Rank		**1**	5	3	4	6	7	2

The bold values indicate the best values.

**Table 13 biomimetics-10-00358-t013:** Comparison of training results.

Index	IPO-MLP	PO-MLP	HHO-MLP	SCA-MLP	OOA-MLP	AOA-MLP	AO-MLP
Best	**1.543 × 10^−2^**	5.685 × 10^−2^	2.703 × 10^−2^	7.628 × 10^−2^	1.634 × 10^−1^	2.037 × 10^−1^	2.381 × 10^−2^
Mean	**4.111 × 10^−2^**	1.069 × 10^−1^	6.166 × 10^−2^	1.919 × 10^−1^	4.586 × 10^−1^	3.488 × 10^−1^	5.172 × 10^−2^
Worst	**3.580 × 10^−1^**	2.471 × 10^−1^	3.486 × 10^−1^	3.945 × 10^−1^	6.099 × 10^−1^	5.521 × 10^−1^	3.560 × 10^−1^
Std	**6.044 × 10^−2^**	5.048 × 10^−2^	6.139 × 10^−2^	7.363 × 10^−2^	1.069 × 10^−1^	9.440 × 10^−2^	5.954 × 10^−2^
Rank	**1**	4	3	5	7	6	2

The bold values indicate the best values.

**Table 14 biomimetics-10-00358-t014:** Comparison of accuracy results (%).

Index	IPO-MLP	PO-MLP	HHO-MLP	SCA-MLP	OOA-MLP	AOA-MLP	AO-MLP
Best	**88.33**	66.67	83.33	71.67	18.33	8.33	85.00
Mean	**70.22**	28.44	44.56	27.67	1.28	0.39	67.39
Worst	**6.67**	0.00	0.00	3.33	0.00	0.00	10.00
Std	**18.32**	23.04	24.85	20.49	3.83	1.56	18.81
Rank	**1**	4	3	5	6	7	2

The bold values indicate the best values.

## Data Availability

Data are available as asked.

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
