# Peer review of "IPO: An Improved Parrot Optimizer for Global Optimization and Multilayer Perceptron Classification Problems"

_biomimetics, 2025, doi:10.3390/biomimetics10060358_

Round 1

Reviewer 1 Report

Comments and Suggestions for Authors

The paper presents a novel optimizer with potential applications. However, major revisions are required to address methodological ambiguities, experimental inconsistencies, and presentation flaws. I recommend "Reconsider after Major Revisions" pending thorough clarification of the improvements, validation of experiments, and correction of formatting issues.

Please check out the attachment for more details.

Author Response

The paper proposes an Improved Parrot Optimizer (IPO) by integrating three strategies: aerial search, modified staying behavior, and roulette fitness-distance balance (RFDB). The IPO is tested on CEC2022 benchmark functions, standard classification datasets, and an oral English teaching quality evaluation problem. While the contributions are promising, significant clarifications and revisions are required to address ambiguities in methodology, inconsistencies in experiments, and presentation issues.

Comments 1: The primary concern lies in the insufficient theoretical justification for the proposed improvements, the motivations and theoretical foundations for the proposed improvements (aerial search, modified staying behavior, RFDB) are unclear. For instance:

  1. The definition and mathematical formulation of RFDB (Section 2.4) are not adequately explained. How does RFDB differ from standard FDB? How is the "roulette wheel selection" integrated into the algorithm?
  2. The rationale for modifying the staying behavior (Equation 17) is vague. Why is the assumption that "the parrot is already on its owner’s body" necessary? How does this enhance local search?
  3. The term Xt RFDB(Equation 18) is undefined. How is this solution selected via RFDB? A step-by-step explanation is missing.

Response 1:

  1. 3.The individual is selected according to the score vector SP[25], which is defined in Equation 16. The higher the score, the greater the chance of being selected. This is a simple selection method based on probability magnitude and is often used in optimization algorithms, such as the well-known ALO algorithm proposed by Mirjalili. Also, for more details, please refer to the reference [25]. Bakır, H. Dynamic fitness-distance balance-based artificial rabbits optimization algorithm to solve optimal power flow problem. Expert Syst. Appl. 2024, 240, 122460.
  2. The staying behavior of PO includes flying to the host and randomly stopping at the host’s body, shown in Equation 6. In the new staying behavior, we remove the flying behavior to consider the case of"the parrot is already on its owner’s body", shown in Equation 17. For each agent, there will be two choices to update their positions. The Equation 17 also will search the space near the current position of agent, thus enhancing the local search.
  3. In Equation 18, we have added the explanation of Xt RFDB, which is ‘Xt RFDBis the selected agent using RFDB selection method.’ And In Section 2.4. Fitness-distance balance (FDB) selection. The Fitness-distance balance (FDB) selection is provided in detailed. For the RFDB, the roulette wheel selection is applied in FDB. The individual is selected according to the score vector SP [25], which is defined in Equation 16. The higher the score, the greater the chance of being selected.

Comments 2: I also have some questions about some of the results of the experiment

  1. The reported results for PO on F1 (1.205E+04 in Table 5 vs. 3.00E+02 in [21]) raise concerns about experimental validity. This inconsistency must be resolved, possibly by re-evaluating PO under identical conditions (e.g., dimensions, parameter settings).
  2. Figure 3 claims IPO outperforms all algorithms on all functions, but Table 6 shows AO ranks higher on F3, F4, F7, and F8. This contradiction needs clarification.
  3. Verify experimental reproducibility, ensure consistency between tables/figures, and discuss discrepancies.

Response 2:

  1. In [21], PO obtains 3.00E+02on F1 of CEC2022 (shown in Table 6). The iterations and population size are set to 30000 and 30. But in our works, the iterations and population size are set to 500 and 30. Therefore, the reported results for POin our works and [21] are obviously different.
  2. Figure 3 shows the convergence curve results of all algorithms in one run. We present the IPO with the best performance. Table 6 presents the statistical results. For a more scientific comparison, the proposed IPO does not have the best statistical performance on all functions, but occasionally achieves the best results.
  3. We have reviewed all the experimental results to ensure that they were reasonable and reproducible.

Comments 3: Furthermore, the writing of many formulas in the article is non-standard or has obvious errors, specifically:

  1. Equation (13): DPiis a scalar, but Equation (15) references normDPi. Clarify whether normalization is applied to DPi.
  2. Equation (15): Use parentheses for clarity (e.g., norm(fi)).
  3. Symbol ≡: Explain its usage in Equations (14) and (16).
  4. Formula Formatting: All "where" clauses should follow equations without line

breaks.

Response 3:

  1. We have modified the Equation (15), where the DPiand fiare modified to DP and f, which are the vector of distance and fitness.
  2. Modifications have been made.
  3. Symbol ≡means this equation is unconditional and holds true under any circumstances.
  4. This suggestion will be modified according to the format of the journal.

Comments 4: Some additional questions:

  1. Table 12 and 13: The interval between the results of AO-MLP and AOA-MLP is too small.
  2. Scientific notation (e.g., 8.833E+01) is inappropriate for classification accuracy percentages (use 88.33% instead).
  3. Figures 3 and 4: Use vector graphics for clarity.
  4. Minor grammatical errors exist (e.g., "the the" in Equation (3) and (11)

description).

  1. Line 20: "well-know optimization algorithms" → "well-known".
  2. Line 25:"classification accuracy of 88.33%, which proves the effectiveness of developed method" → "the developed method".
  3. Line 62: "resulting inaccurate evaluation results" → "resulting in inaccurate evaluation results".
  4. In Figure 5: The first letter should be capitalized instead, "English teaching quality evaluation based on principal component analysis and support vector machine" → "English Teaching Quality Evaluation Based on Principal Component Analysis and Support Vector Machine"

The paper presents a novel optimizer with potential applications. However, major revisions are required to address methodological ambiguities, experimental inconsistencies, and presentation flaws. I recommend "Reconsider after Major Revisions" pending thorough clarification of the improvements, validation of experiments, and correction of formatting issues.

Response 4: Thanks to this series of detailed suggestions, we have refined them one by one.

Reviewer 2 Report

Comments and Suggestions for Authors

The manuscript introduces a new modification of the Parrot Optimizer, which integrates multiple improvements to solve global optimization problems. The results indicate superior performance over several state-of-the-art algorithms using multiple real-world classification datasets.

COMMENTS:

  1. Wilcoxon signed-rank test is applied, however, confidence intervals or effect sizes are not reported, which would strengthen the claim of statistical significance.
  2. A table summarizing the computational time for each algorithm on benchmark functions should be added.
  3. The algorithm’s theoretical convergence properties or complexity beyond big-O notation should not discussed. 

Author Response

The manuscript introduces a new modification of the Parrot Optimizer, which integrates multiple improvements to solve global optimization problems. The results indicate superior performance over several state-of-the-art algorithms using multiple real-world classification datasets.

Comments 1: Wilcoxon signed-rank test is applied, however, confidence intervals or effect sizes are not reported, which would strengthen the claim of statistical significance.

Response 1: Agree. Thank you for pointing this out. In the revised manuscript, We have added the description of the number of IPO results that are compared with those of the comparative algorithms, which is as follows: In this test, The optimization results of IPO are compared respectively with those of each algorithm. The number of the comparison results is 15.

Comments 2: A table summarizing the computational time for each algorithm on benchmark functions should be added.

Response 2: Agree. We have added a table (Table 9) of computational time in the revised manuscript. The result analysis is as follows: Table 9 reports the results of average computational time on each test function. It can be seen that due to the addition of several improvement strategies, the calculation time of IPO is slightly longer than that of PO. The AOA algorithm has the shortest calculation time because of its simple structure, while the AO algorithm is more complex and has the longest calculation time.

Comments 3: The algorithm’s theoretical convergence properties or complexity beyond big-O notation should not discussed.

Response 3: Thanks a lot for your comment. We have added it as it is required by the other 2 reviewers.

Reviewer 3 Report

Comments and Suggestions for Authors

In this study, the authors proposed IPO, an improved version of the PO algorithm inspired by parrot behavior. However, the study contains significant limitations in both methodological and practical aspects. Therefore, further development of the scope, implementation, and presentation of the study is essential. Therefore, the article should be rejected at this stage.

1) The original aspects of the study should be clearly explained in the abstract.

2) The literature section should be discussed in more detail.

3) The connection between oral English education and the optimization algorithm should be established more clearly.

4) The writing of the equations should be standard.

5) There are too many formulas, but very few of these formulas have explanations.

6) The “empirical formula” can be detailed in the MLP structure section. (For example, Why was 2n+1 chosen?)
7) The boxplot graph is not readable. Improve the colors.

8) Ablation test results should be supported by statistical analysis.
9) For each function type, such as unimodal, multimodal..., the performance of IPO should be evaluated separately.

10) Dataset features should be presented in more detail.

11) While MSE is low for datasets, classification accuracy should be high, right? Do the results support this?

12) Why was MLP used for 60 samples? Why not k-NN, Decision Tree, SVM,..

13) Future work suggestion is weak.

Author Response

In this study, the authors proposed IPO, an improved version of the PO algorithm inspired by parrot behavior. However, the study contains significant limitations in both methodological and practical aspects. Therefore, further development of the scope, implementation, and presentation of the study is essential. Therefore, the article should be rejected at this stage.

Comments 1: The original aspects of the study should be clearly explained in the abstract.

Response 1: Agree. We have added the details of original aspects in the abstract, which are: The aerial search strategy is derived from Arctic Puffin Optimization and is employed to enhance the exploration ability of PO. The staying behavior and communicating behavior of PO are modified using random movement and RFDB selection method to achieve a better balance between exploration and exploitation.

Comments 2: The literature section should be discussed in more detail.

Response 2: Thanks a lot for your valuable comments. We have updated the literature section.

Comments 3: The connection between oral English education and the optimization algorithm should be established more clearly.

Response 3: In this works, the oral English education evaluation problem is regarded as a multi-layer perceptron classification problem. And the related descriptions are also modified for better understanding, which are:

4.3. Case 3: Oral English Education Evaluation Problem

In the third case, the proposed IPO is used to solve the oral English education evaluation problem. The oral English education evaluation problem can be regarded as a classification problem with multiple features. In this paper, the evaluation model is constructed using a multilayer perceptron model with 10-21-3 structure. Then the weights and biases of this MLP model is optimized by using the proposed IPO. The results are shown in below.

4.3.1. Indexes of oral English teaching quality evaluation model

For better evaluating the oral English teaching quality, the indexes of oral English teaching quality evaluation problem is selected according to the works in [19]. The index system is constructed as shown in Figure 5, which has five first-grade indexes: qualities of teacher, teaching attitude, teaching content, teaching method, and teaching effectiveness. In each first-grade index, there are one or more second-grade indexes. These elements play a leading role in the oral English teaching evaluation and ensure the scientific and reasonable teaching quality evaluation system.

As can be seen in Figure 5, ten features in the oral English education evaluation problem are selected to determine the evaluation outcomes of teachers. The evaluation results are be divided into three cases, which are excellent, good and qualified. Therefore, a MLP model with 10-21-3 structure is constructed to find the relationship between the indexes and evaluation outcomes. The node number of hidden layer is determined by a empirical formula 2×n+1 [33], where n is the the number of input parameters.

Comments 4: The writing of the equations should be standard.

Response 4: Thank you for this suggestion. We have carefully checked for all equations and revised some inappropriate expressions, such as the equations (3) and (10).

Comments 5: There are too many formulas, but very few of these formulas have explanations.

Response 5: Agree. We have added some explanations for these formulas.

For Equation (1), the explanation is: By using the Equation (1), the initial positions of population individuals are randomly generated within the upper and lower boundary ranges.

For Equation (6), the explanation is: Xbest×levy(D) represents the behavior of flying to the owner and rand×ones(1, D) represents the behavior of staying on the owner's body for a while.

For Equations (8)-(10), the explanation is: where O denotes the behavior of flying towards the owner, and L denotes the behavior of moving away from the strangers.

Comments 6: The “empirical formula” can be detailed in the MLP structure section. (For example, Why was 2n+1 chosen?)

Response 6: The MLP structure used in this study is 10-21-3, where 10 is the number of input features and 3 is the number of output classes. The hidden layer size (21 neurons) was selected based on the empirical formula 2n+1, where n=10 to provide a balance between learning capacity and generalization, as commonly recommended in neural network design heuristics.

Comments 7: The boxplot graph is not readable. Improve the colors.

Response 7: Thank you for this suggestion. We have beautified the box diagram, as shown in Figure 4. The box plot color of each algorithm is consistent with the convergence curve.

Comments 8: Ablation test results should be supported by statistical analysis.

Response 8: To prove the power of all strategies together, we preform a Wilcoxon signed-rank test and from the results, it can be confirmed that the observed the performance differences are significant (p < 0.05).

Comments 9: For each function type, such as unimodal, multimodal..., the performance of IPO should be evaluated separately.

Response 9: We appreciate the reviewer’s suggestion. However, presenting separate evaluations for each function type (e.g., unimodal, multimodal, etc.) would lead to a significant amount of redundant text and multiple sub-subsections, which may disrupt the flow of the paper. Therefore, we opted for a unified comparison to maintain clarity and conciseness, while placing greater emphasis on the main contribution — the application of IPO to the multilayer perceptron classification problem.

Comments 10: Dataset features should be presented in more detail.

Response 10: In the revised paper, we have added more details about the dataset, such as: The index system is constructed as shown in Figure 5, which has five first-grade indexes: qualities of teacher, teaching attitude, teaching content, teaching method, and teaching effectiveness. In each first-grade index, there are one or more second-grade indexes.

Comments 11: While MSE is low for datasets, classification accuracy should be high, right? Do the results support this?

Response 11: Yes it is only in scientific notation.

Comments 12: Why was MLP used for 60 samples? Why not k-NN, Decision Tree, SVM,..

Response 12: In this work, we mainly studied the improvement of the PO algorithm and applied it to the optimization training of the parameters of the multi-layer perceptron. In the application problem, in addition to testing five standard classification datasets, we also studied the oral English teaching quality evaluation problem. Models such as k-NN, Decision Tree, and SVM can also be used to solve these classification problems. We will consider the effects of different models on these classification problems in the next step of work.

Comments 13: Future work suggestion is weak.

Response 13: We have rewrite it. This is the new one

“In future work, the suggested optimizer can be extended to tackle a broader range of complex optimization problems, such as robotic path planning, feature selection in high-dimensional datasets, and dynamic scheduling tasks. Additionally, the IPO-MLP framework can be integrated with other metaheuristic algorithms to enhance convergence speed and solution quality. Another promising direction is the automatic optimization of MLP architecture, including activation function selection, hidden layer configuration, and neuron count, potentially using self-adaptive or co-evolutionary strategies. These enhancements can further improve the robustness and generalization ability of the proposed approach across diverse application domains.”

Reviewer 4 Report

Comments and Suggestions for Authors

The article seems quite interesting. The authors propose in the article an improvement of the metaheuristic algorithm called “Parrot Optimizer (PO),” which belongs to the field of so-called bioinspired algorithms. The new improved model, called “Improved Parrot Optimizer” (IPO), incorporates new and innovative strategies such as “aerial search,” “modified staying behavior,” and “improved communicating behavior.” These improvements add more value and capabilities to the algorithm. Moreover, the authors have conducted a validation by presenting an extensive evaluation with the CEC2022 benchmark functions and several classification dataset packages. The paper includes statistical analyses and comparisons (such as the Wilcoxon test and Friedman ranking), which enhance the credibility and robustness of the results obtained. They also show a practical application by evaluating the IPO in a real educational context (assessment of oral English teaching), demonstrating the algorithm’s versatility and applicability. However, there are some aspects and sections of the article that could be improved and expanded. For example, in terms of language, there are some grammatical and stylistic errors that affect clarity, especially in the abstract and the introduction (for instance when it says: “the IPO has been applied to solve the oral English teaching quality evaluation problem”). My simple recommendation is that the authors conduct a new linguistic revision to improve the quality of the scientific and academic language. Another aspect that could significantly improve the article is a more thorough expansion and critical review of previous works on this topic, especially regarding the state of the art in optimization based on “animal behavior.” In this regard, the discussion or analysis could be expanded to explain why and how parrot behavior, in this case, justifies specific improvements in exploration/exploitation. Similarly, although many numerical results are presented, there is a lack of discussion and commentary on the interpretability of why the IPO outperforms other current methods. A more analytical section would be useful, delving deeper into the reasons behind IPO’s superior performance. The authors have applied this improved algorithm to an educational application, but it seems underdeveloped. That is, although the application to the evaluation of oral English teaching is mentioned, the description of the dataset package, the classification criteria, and the educational implications are superficial. The authors could provide more detailed information on these three aspects. In this sense, it would be advisable for the authors to expand the case study description and critically discuss its educational relevance in greater depth. These improvements would give the article a bit more weight for its potential publication.

Author Response

The article seems quite interesting. The authors propose in the article an improvement of the metaheuristic algorithm called “Parrot Optimizer (PO),” which belongs to the field of so-called bioinspired algorithms. The new improved model, called “Improved Parrot Optimizer” (IPO), incorporates new and innovative strategies such as “aerial search,” “modified staying behavior,” and “improved communicating behavior.” These improvements add more value and capabilities to the algorithm. Moreover, the authors have conducted a validation by presenting an extensive evaluation with the CEC2022 benchmark functions and several classification dataset packages. The paper includes statistical analyses and comparisons (such as the Wilcoxon test and Friedman ranking), which enhance the credibility and robustness of the results obtained. They also show a practical application by evaluating the IPO in a real educational context (assessment of oral English teaching), demonstrating the algorithm’s versatility and applicability. However, there are some aspects and sections of the article that could be improved and expanded.

Comments 1: For example, in terms of language, there are some grammatical and stylistic errors that affect clarity, especially in the abstract and the introduction (for instance when it says: “the IPO has been applied to solve the oral English teaching quality evaluation problem”). My simple recommendation is that the authors conduct a new linguistic revision to improve the quality of the scientific and academic language.

Response 1: We appreciate the reviewer’s feedback regarding the language quality. A thorough linguistic revision has been conducted throughout the manuscript, with particular attention to the abstract and introduction. Regarding the phrase in question, we clarify that the oral English teaching quality evaluation problem was used as a dataset and formulated as a classification task, which was addressed using a multilayer perceptron model optimized by IPO. This has now been rephrased to improve clarity and reflect the intended meaning.

Comments 2: Another aspect that could significantly improve the article is a more thorough expansion and critical review of previous works on this topic, especially regarding the state of the art in optimization based on “animal behavior.” In this regard, the discussion or analysis could be expanded to explain why and how parrot behavior, in this case, justifies specific improvements in exploration/exploitation.

Response 2: We appreciate the reviewer’s valuable suggestion. In response, we would like to clarify that Table 1 of the manuscript provides a selection of relevant studies from 2018 to 2023 that apply various bio-inspired algorithms, including Genetic Algorithm (GA), Improved Quantum Particle Swarm Optimization (QPSO), and the Improved Crow Search Algorithm (CSA), to related optimization problems. These examples help highlight the relevance and evolution of animal behavior-based metaheuristics. Additionally, we have expanded the discussion in the revised manuscript to more clearly articulate how the modeled parrot behavior contributes to exploration and exploitation dynamics in IPO, emphasizing its novelty compared to existing approaches.

Comments 3: The authors have applied this improved algorithm to an educational application, but it seems underdeveloped. That is, although the application to the evaluation of oral English teaching is mentioned, the description of the dataset package, the classification criteria, and the educational implications are superficial. The authors could provide more detailed information on these three aspects. In this sense, it would be advisable for the authors to expand the case study description and critically discuss its educational relevance in greater depth. These improvements would give the article a bit more weight for its potential publication.

Response 3: Agree. In the revised paper, we have modified the Section 4.3, as shown in follows:

4.3. Case 3: Oral English Education Evaluation Problem

In the third case, the proposed IPO is used to solve the oral English education evaluation problem. The oral English education evaluation problem can be regarded as a classification problem with multiple features. In this paper, the evaluation model is constructed using a multilayer perceptron model with 10-21-3 structure. Then the weights and biases of this MLP model is optimized by using the proposed IPO. The results are shown in below.

4.3.1. Indexes of oral English teaching quality evaluation model

For better evaluating the oral English teaching quality, the indexes of oral English teaching quality evaluation problem is selected according to the works in [19]. The index system is constructed as shown in Figure 5, which has five first-grade indexes: qualities of teacher, teaching attitude, teaching content, teaching method, and teaching effectiveness. In each first-grade index, there are one or more second-grade indexes. These elements play a leading role in the oral English teaching evaluation and ensure the scientific and reasonable teaching quality evaluation system.

As can be seen in Figure 5, ten features in the oral English education evaluation problem are selected to determine the evaluation outcomes of teachers. The evaluation results are be divided into three cases, which are excellent, good and qualified. Therefore, a MLP model with 10-21-3 structure is constructed to find the relationship between the indexes and evaluation outcomes. The node number of hidden layer is determined by a empirical formula 2×n+1 [33], where n is the the number of input parameters.

More, we added the description of educational implication in Section 4.3.2: By using the proposed IPO-MLP, the decision-makers can better allocate teaching resources based on the accurate assessment of teachers' teaching levels.

Round 2

Reviewer 1 Report

Comments and Suggestions for Authors

Comments on the Quality of English Language

The overall English is generally clear and understandable. However, there are still minor issues related to grammar, word choice, and formatting that need to be addressed for improved readability and professionalism.

Author Response

Comments 1: I appreciate the authors’ efforts in revising the manuscript and addressing the concerns raised in the first-round review. However, after carefully evaluating the revised manuscript and the authors’ responses, I find that many of the key concerns remain insufficiently addressed.

One of my primary concerns in the first-round review was the lack of theoretical or empirical rationale for the proposed modifications to the Parrot Optimizer (i.e., aerial search, modified staying behavior, and RFDB). Unfortunately, the authors' responses mostly restate the original descriptions without offering deeper insights. Without clear justification, these modifications appear as arbitrary combinations rather than informed improvements.

Response 1: The Parrot Optimizer (PO) is a newly proposed optimization algorithm based on the behaviors of trained Pyrrhura Molinae parrots. To enhance the searching abilities of PO, we have applied three methods: i.e., aerial search strategy, new staying behavior, and new communicating behavior. In our works, we find that the aerial search strategy of Arctic Puffin Optimization (APO) has shown strong global exploration. Thus this strategy has been adopted for the IPO. Moreover, in the Staying Behavior of PO, there is only one equation to update the position. According to the staying behavior characteristics of Parrot, we assume that the parrot is already on its owner’s body and might move randomly. This case is reasonable for the staying behavior of Parrot and also will increase the local search of PO. For the third modification, we introduce the RFDB selection in the Communicating Behavior to balance the global and local exploration of PO. To sum up, these modifications are determined based on the shortcomings of PO. Thank you for this insight suggestions and we also have added more detailed descriptions in the paper.

Comments 2: âš« The RFDB selection mechanism, described in Section 2.4, is still unclear. How the roulette wheel selection is integrated with fitness-distance balance remains vague. The response refers to another paper [25], but this crucial mechanism must be self-contained and clearly explained in this manuscript. âš« Equation (18) introduces the term Xt RFDB but its selection process via the RFDB method is still not fully described. A step-by-step explanation or illustrative pseudocode is necessary.

Response 2: We have added the descriptions of roulette wheel selection, which is: In the RFDB, the scores of all individuals are summed, which is Ssum=S1+S2+…+SN. Then the probability of each individual being selected is the ratio of the individual's score to the sum of all scores, which is Si/Ssum. Therefore, the higher the score, the greater the chance of being selected. 

Comments 3: In the second round I saw that the author added the results of the algorithm's running time, but it seems to be inconsistent with the results of the complexity analysis. The reported computation time (Table 9) contradicts the authors’ own statement in the computational complexity analysis that IPO and PO share the same complexity. IPO consistently takes more time than PO in Table 9. This discrepancy must be clarified, and the hardware configuration used for experiments should be reported.

Response 3: In Table 9, the IPO has shown slightly longer calculation time than those of PO. Due to the fact that IPO has more options and calculation formulas, the result is reasonable. In the complexity analysis, we only consider the computational complexity updating position. In this case, IPO and PO have the same complexity.

The hardware configuration is Intel(R) Core(TM) i7-10700 CPU @ 2.90 GHz and32.00 GB RAM, which has been added in the Section 4. Experimental Results.

Comments 4: Several formatting and grammatical issues remain unresolved or introduced:

âš« RFDB is used in the abstract without full expansion. Provide the full term upon first use.

âš« Inconsistent use of acronym "PO" vs. "Parrot Optimizer". Please unify terminology throughout the text.

âš« Minor formatting issues such as:

â—¼ Table 4: no spacing between columns in the first row.

â—¼ Table 9: columns for IPO and PO are too close.

â—¼ Line 25: “well-know” (I've already mentioned the misspelling of the word in my first review).

â—¼ Line 283: missing space before “and”.

â—¼ Line 256: avoid table headers breaking across pages.

Response 4:

RFDB is replaced by roulette fitness-distance balance selection method in the abstract.

Parrot Optimizer is replaced by PO in the paper.

Tables 4 and 9 are modified.

“well-know” is modified into “well-known”

Line 295: space is added before “and”.

The problem of table headers breaking across pages will be solved during the proofreading process.

Reviewer 2 Report

Comments and Suggestions for Authors

The paper has been significantly improved. 

Author Response

Thank you very much for your approval.

Reviewer 3 Report

Comments and Suggestions for Authors

The authors generally made all revisions. However, they should also take into account Comment 12. Different ML results should be visible in the article.

Author Response

Comments 1: The authors generally made all revisions. However, they should also take into account Comment 12. Different ML results should be visible in the article.

Response 1: We are sorry that we were unable to make the modifications in this regard. Our main work has been to improve the PO algorithm and apply it to solve global optimization problems and MLP classification problems. We understand the functions of other machine learning models such as SVM, KNN and decision trees. We will apply and compare them in our future work.

Reviewer 4 Report

Comments and Suggestions for Authors

The authors have made an effort to improve the paper in the areas highlighted by the previous review. The paper is now better justified for publication.

Author Response

Comments 1: The authors have made an effort to improve the paper in the areas highlighted by the previous review. The paper is now better justified for publication.

Response 1: Thank you very much for your valuable suggestions and opinions.